# Gender differences in provider practice characteristics and medicare payment & services among diagnostic radiologists

Ajay Malhotra[1]*, Chris J. Lee[1], Mihir Khunte[1], Raj Moily[1], Dheeman Futela[1], Seyedmehdi Payabvash[2], David Seidenwurm[3], Dheeraj Gandhi[4]

1 Department of Radiology and Biomedical Imaging, Yale School of Medicine, New Haven, Connecticut, United States of America, 2 Department of Radiology, Columbia University Medical Center, New York, New York, United States of America, 3 Radiology, Sutter Health, Sacramento, California, United States of America, 4 Radiology, Neurology and Neurosurgery, University of Maryland School of Medicine, Baltimore, Maryland, United States of America

* ajay.malhotra@yale.edu

## Abstract

### Background

Gender-based differences in representation, practice settings, sub-specialty, billings, and payments among radiologists are poorly understood.

### Purpose

To compare representation, scope of practice, productivity, and Medicare payments between female and male diagnostic radiologists.

### Methods and materials

This cross-sectional (2017–2021) retrospective study examined the CMS Medicare Fee-for-Service Provider Utilization and Payment Database for payments to male and female diagnostic radiologists. The study practice setting, volume of services provided, number of unique beneficiaries served and financial data. Descriptive statistics were reported as mean with standard error (SE) for financial variables and median with interquartile range (IQR) for non-financial variables.

### Results

Between 2017–2021 data, a total of 33,029 diagnostic radiologists provided services to Medicare fee-for-service beneficiaries, of whom 8,217 (24.9%) were female. Female radiologists were disproportionately in academic versus non-academic radiology (34.2% vs 22.6%); practicing in urban versus rural areas (26.3% vs 17.0%); in larger practices (30.1% in practices with ≥100 radiologists vs 17.5% in practices with 1–9 radiologists); and in certain subspecialties (breast imaging- 64.9%). On average, female radiologists received 86% of the total Medicare payments received by male radiologists (mean

**Data availability statement:** DATA AVAILABILITY STATEMENT This study used publicly available data from the CMS Medicare Physician & Other Practitioners dataset (2017–2021). The data are accessible at: https://data.cms.gov/provider-summary-by-type-of-service/medicare-physician-other-practitioners-by-provider-and-service. The data was accessed in October 2024.

**Funding:** The author(s) received no specific funding for this work.

**Competing interests:** The authors have declared that no competing interests exist.

[SE], $301,931 [$1810] vs $352,604 [$1,164]; p<.001). Female radiologists billed fewer median total services (6,660 vs 9,859; p<.001), served fewer beneficiaries (5,441 vs 7,678; p<.001), and billed fewer unique codes (20 vs 39; p<.001). Overall, female radiologists' average (SE) payment per service was 18% higher than males [$36.26 (0.13) versus $30.71 (0.06), respectively, but were similar within each subspecialty.

## Conclusion

Female radiologists are disproportionately in academic, urban, and large practices and work as subspecialists. Further, while total Medicare payments received by females were less than males, average payments per service were higher for females overall, but comparable by sex in the same subspecialty.

## Introduction

Female representation in the radiology workforce is less than for males and has only marginally increased in the last decade [1–3]. Financial compensation is important for physician recruitment and retention, and recent studies have shown differences in salaries between females and males in academic radiology [4–7]. However, the gender-based differences in representation, practice settings, sub-specialty, billings, and payments in radiology practice remain poorly understood.

Since 2010, the Centers for Medicare & Medicaid Services (CMS) has published data on physician demographics and reimbursements that is publicly available [8]. These data allow for a thorough analysis of billing and payments for physicians in different practice settings throughout the United States. Previous studies using these data have shown the existence of a gender pay gap for physicians overall and in specialties such as, but not limited to, radiation oncology, ophthalmology, otolaryngologists, cardiology, and neurosurgery [9–13]. Female physicians are estimated to earn approximately $2 million less than their male counterparts over a 40-year career even when controlling for the number of hours worked, clinical volume, and specialty [14]. This trend has been demonstrated in studies focusing on different physician subpopulations across different specialties, different practice types, as well as geographic locations [15]. However, these differences for female and male U.S. radiologists have not been well described.

As pay differences between genders are associated with various factors, the objective of this study is to assess any gender-based differences in radiologists' practice volume (number of charged services and beneficiaries), breadth of practice (number of unique Medicare billing codes), charges and payments (the mean dollar amount Medicare was charged and the payment received from Medicare) by analyzing the publicly available CMS data over a 5-year period (2017–2021).

## Methods

### Data

This cross-sectional retrospective study examined the CMS Medicare Fee-for-Service Provider Utilization and Payment Database for payments to diagnostic radiologists

from January 1, 2017, to December 31, 2021 [8]. As the data are publicly available, the study was not subject to institutional review board review oversight. The data was accessed in October 2024. Physician-level payment information was obtained for services performed and their corresponding Medicare reimbursements from 2017 to 2021. Each row of the dataset contains a unique combination of Current Procedural Terminology (CPT) or Healthcare Common Procedure Coding System (HCPCS) codes and National Provider Identifier (NPI) allowing identification of services provided by individual physicians. These data were supplemented with physician demographic information, practice location, number of services rendered, number of beneficiaries served, and Medicare payments and charges. Information on the number of years in practice was derived from the physician's medical school graduation year obtained from the publicly available National Plan & Provider Enumeration System's database and linked using the clinician's NPI [16]. The Neiman Imaging Types of Service (NITOS) database was used to categorize CPT/HCPCS codes based on characteristics such as invasiveness, modality, body region, and focus [17]. Specifically, NITOS was used to categorize the modality and body region of the imaging for each service rendered by radiologists. The unit of analysis was at the individual physician level. Physicians were included in the study if their self-reported specialty code for Medicare was diagnostic radiology, and they practiced in the 50 states or District of Columbia.

## Variables

The primary variable of interest was gender, which was a self-reported binary variable obtained from the National Plan & Provider Enumeration System's database. Diagnostic radiologists were classified by gender to examine differences in clinical practice characteristics and Medicare-related financial outcomes. First, clinical practice volume was assessed by the total number of services provided and the number of unique beneficiaries served. To characterize the breadth of practice for each physician, we identified the number of unique CPT/HCPCS codes billed. Financial variables included were charges submitted to Medicare and payments received from Medicare. All financial data were adjusted for inflation to 2021 values using the Consumer Price Index [18].

Radiologists were categorized as either generalists or in one of five subspecialties: abdominal, breast, cardiothoracic, musculoskeletal, and neuroradiology. Subspecialty identification was based on work relative value units (wRVUs) linked to claims in the CMS "Medicare Physician & Other Practitioners – by Provider and Service" database. Each claim code was categorized into one of the five subspecialties using imaging modality and body region according to the NITOS system. This was necessary because CMS specialty data simply identify diagnostic radiologists. They do not indicate the radiologic subspecialty. Each radiologist was identified as practicing within a certain subspecialty if more than 50% of their total wRVUs were in that subspecialty. Those not meeting the 50% threshold for a subspecialty were categorized as generalists. This threshold and the NITOS-based classification system have been validated in both the academic and private practice settings in previous studies and sensitivity analyses as a robust approach for subspecialty identification [19–21]. Radiologists subspecializing in nuclear medicine and vascular & interventional radiology were excluded from the dataset to focus exclusively on diagnostic radiologists. Academic and non-academic status was determined using data from the Harvey L. Neiman Health Policy Institute Academic Radiology Practices categorized the academic status of practices associated with radiologists in CMS files [22]. Urban and rural practice locations were determined using Rural-Urban Commuting Area (RUCA) codes based on the practice ZIP code [23]. Practice size was defined as the number of physicians working in the same group, which is available within the CMS data.

## Statistical analyses

All analyses were conducted with R version 4.4.2 (R Foundation for Statistical Computing, Vienna, Austria). All statistical tests were two-tailed, with an alpha level of 0.05 used to determine significance. Trends in payment amounts over time were determined with a time-series analysis with a two-tailed alpha level (α = 0.05). Descriptive statistics were reported as

mean with standard error (SE) for parametric data (e.g., charges, payments) and median with interquartile range (IQR) for nonparametric data (e.g., number of services, beneficiaries, and codes). Independent t-test were used to compare mean values while the Mann-Whitney U test (also known as the Wilcoxon t-test) was used to compare medians.

## Results

The 2017–2021 data included a total of 33,029 diagnostic radiologists who provided services to Medicare beneficiaries, of whom 8,217 (24.9%) were female and 24,812 (75.1%) were male. The number of female radiologists increased from 6,747 in 2017–7,321 in 2021 (p < .001), and the number of male radiologists increased from 21,033 in 2017–21,610 in 2021 (p = .003). Hence, over the 2017–2021 period, the female representation increased 1 percentage point from 24.3% to 25.3% of all radiologists (Table 1).

Relative to their overall mean representation among diagnostic radiologists (24.9%), 34.2% of academic radiologists were female, compared to 22.6% in non-academic settings; The proportion of female radiologists also varied by practice location and size: 26.3% practiced in urban areas versus 17.0% in rural areas, and 30.1% were in large practices (≥100 radiologists) compared to 17.5% in small practices (1–9 radiologists). Subspecialty distribution showed the highest female representation particularly in breast imaging, where 64.9% of practitioners were female, followed by cardiothoracic (29.7%) and abdominal imaging (29.4%), while lower proportions were observed in neuroradiology (21.7%), musculoskeletal imaging (14.9%), and general radiology (20.6%) (Table 1). By years of practice, female radiologists had disproportionately higher representation with 10–24 years (28.0%) or 1–9 years (25.5%) versus those with ≥25 years of practice (22.2%). The Northeast region had the highest share of female radiologists (31.6%). Geographically, there was variation between states. The highest female representation was seen in the Northeast, near Washington D.C., California, and Washington state (Fig 1).

On average, female radiologists submitted charges annually that were 76% of that submitted by males (mean [SE], $1,472,306 [$9,066] vs $1,928,044 [$6,363]; p < .001) and were paid by Medicare 86% of what males were (mean [SE], $301,931 [$1810] vs $352,604 [$1,164]; p < .001). The total number of annual services that female billed were 68% of the services billed by males (6,660 [IQR, 3,168–11,990] vs 9,859 [IQR, 4,938–16,575]; p < .001). Females annually served 71% as many beneficiaries (5,441 [IQR, 2,694–9,159] vs 7,678 [IQR, 3,958–12,231]; p < .001), and billed 51% as many unique codes (20 [IQR, 10–37] vs 39 [IQR, 21–55]; p < .001) (Table 2). Hence, normalizing payments based on services rendered, female radiologists received an average (SE) payment of $36.26 (0.13) that is 18% higher than the $30.71 (0.06) average payment per service rendered by male radiologists (Table 3).

Table 3 shows similar patterns by the characteristics of the radiologists. Female academic radiologists' payments from Medicare were 89% of academic males ($214,777 vs $241 078). The results show similar patterns for urban/rural, years of practice, practice size, region, and subspecialty. For example, females received lower total payments than males for every subspecialty. The smallest female-to-male payment ratio was 75% for musculoskeletal imaging ($208,936 for females vs $277,424 for males; p < 0.001) and the largest ratio was 93% for cardiothoracic imaging ($153,356 vs $142,146; p = 0.016). Normalizing these payments based on services rendered, Table 3 also shows the average payments per service for female and male radiologists by their characteristics. Female compared with male average payments per service are 7% higher for academic radiologists ($32.99 vs $30.96) and 23% higher for non-academic radiologists. Likewise, average payments per service are higher for female than male radiologists in both urban (17%) and rural areas (8%) and regardless of years of practice, practice size, or region of the county. In contrast, there is generally parity between female and male radiologists in payments per service for those in the same subspecialty, ranging from 6% less for musculoskeletal to 5% more for generalists.

To better understand whether the observed gender differences persisted after accounting for practice and workload variables, we performed a multivariable linear regression analysis. The model adjusted for the number of services provided, number of beneficiaries, rural versus urban practice location, years in practice, group size, geographic region, and subspecialty. After adjustment, female radiologists remained significantly associated with lower total annual submitted

**Table 1. Demographics and percentage representation of women in radiology.**

| | Percent female % (No./Total no.) | | |
|---|---|---|---|
| | **2017** | **2021** | **All years** |
| **Overall** | 24.3 (6,747/27,780) | 25.3 (7,321/28,931) | 24.9 (8,217/33,029) |
| **Academic status** | | | |
| Academic | 34.5 (1,619/4,692) | 34.3 (1,983/5,774) | 34.2 (221/948) |
| Nonacademic | 22.2 (3,380/15,232) | 22.6 (3,975/17,619) | 22.6 (176/1,131) |
| **Practice rurality** | | | |
| Urban | 25.8 (5,146/19,909) | 26.3 (6,194/23,559) | 26.3 (425/2,199) |
| Rural | 16.7 (476/2,843) | 17.0 (547/3,216) | 17.0 (32/182) |
| **Years in practice** | | | |
| <10 | 24.9 (101/405) | 25.6 (934/3,651) | 25.5 (59/365) |
| 10-24 | 27.5 (3,242/11,804) | 28.0 (3,505/12,535) | 28.0 (260/1,239) |
| ≥25 | 22.0 (2,420/10,997) | 22.2 (2,467/11,126) | 22.2 (147/806) |
| **Group practice size** | | | |
| <10 members | 16.9 (275/1,625) | 17.3 (312/1,799) | 17.5 (4/36) |
| 10–49 member | 19.5 (1,246/6,380) | 19.7 (1,450/7,344) | 19.8 (51/284) |
| 50–99 members | 22.9 (813/3,552) | 23.3 (981/4,207) | 23.1 (60/425) |
| ≥100 members | 29.7 (3,363/11,317) | 30.1 (4,100/13,630) | 30.1 (352/1,663) |
| **Geographic region** | | | |
| Midwest | 22.0 (1,355/6,167) | 23.2 (1,585/6,830) | 22.3 (108/608) |
| Northeast | 30.9 (1,952/6,309) | 32.0 (2,049/6,399) | 31.6 (164/688) |
| South | 21.7 (2,034/9,374) | 22.8 (2,209/9,695) | 22.4 (149/896) |
| West | 23.5 (1,350/5,742) | 24.4 (1,420/5,811) | 24.2 (105/509) |
| **Subspecialty** | | | |
| Abdominal | 29.7 (969/3,265) | 29.6 (1,077/3,634) | 29.4 (1,203/4,088) |
| Breast | 64.2 (2,345/3,654) | 67.9 (2,498/3,681) | 64.9 (2,749/4,236) |
| Cardiothoracic | 28.8 (373/1,295) | 31.1 (442/1,421) | 29.7 (571/1,923) |
| Musculoskeletal | 20.9 (339/1,621) | 21.5 (373/1,734) | 21.7 (430/1,983) |
| Neuroradiology | 19.6 (451/2,298) | 20.3 (534/2,631) | 20.6 (603/2,933) |
| Generalists | 14.5 (2,270/15,647) | 15.1 (2,397/15,830) | 14.9 (2,661/17,866) |

charges (p = 0.004). However, they were also associated with higher total Medicare payments, independent of clinical volume and practice characteristics (p = 0.001). These findings suggest that female radiologists, on average, may provide a mix of services that are reimbursed at higher rates, potentially reflecting differences in procedural focus, coding strategies, or patient complexity not captured by raw service counts.

For the entire 2017–2021 period, Medicare payments to female radiologists were 86% that of males. However, over this period, this ratio trended upward from 83% in 2017 to 89% in 2021 (Fig 2). At the same time, the ratio of female-to-male services rendered also increased from 64% in 2017 to 72% in 2021, and the ratio of female-to-male payments per service rendered was 115% in 2017 and 116% in 2021. The combination of the constant parity per service and increasing number of services are associated with the improving aggregate Medicare payment parity between female and male radiologists.

## Discussion

This retrospective analysis of gender-based differences among diagnostic radiologists found that relative to the one-quarter of radiologists who are female, female radiologists are disproportionately in academic settings, urban, and large

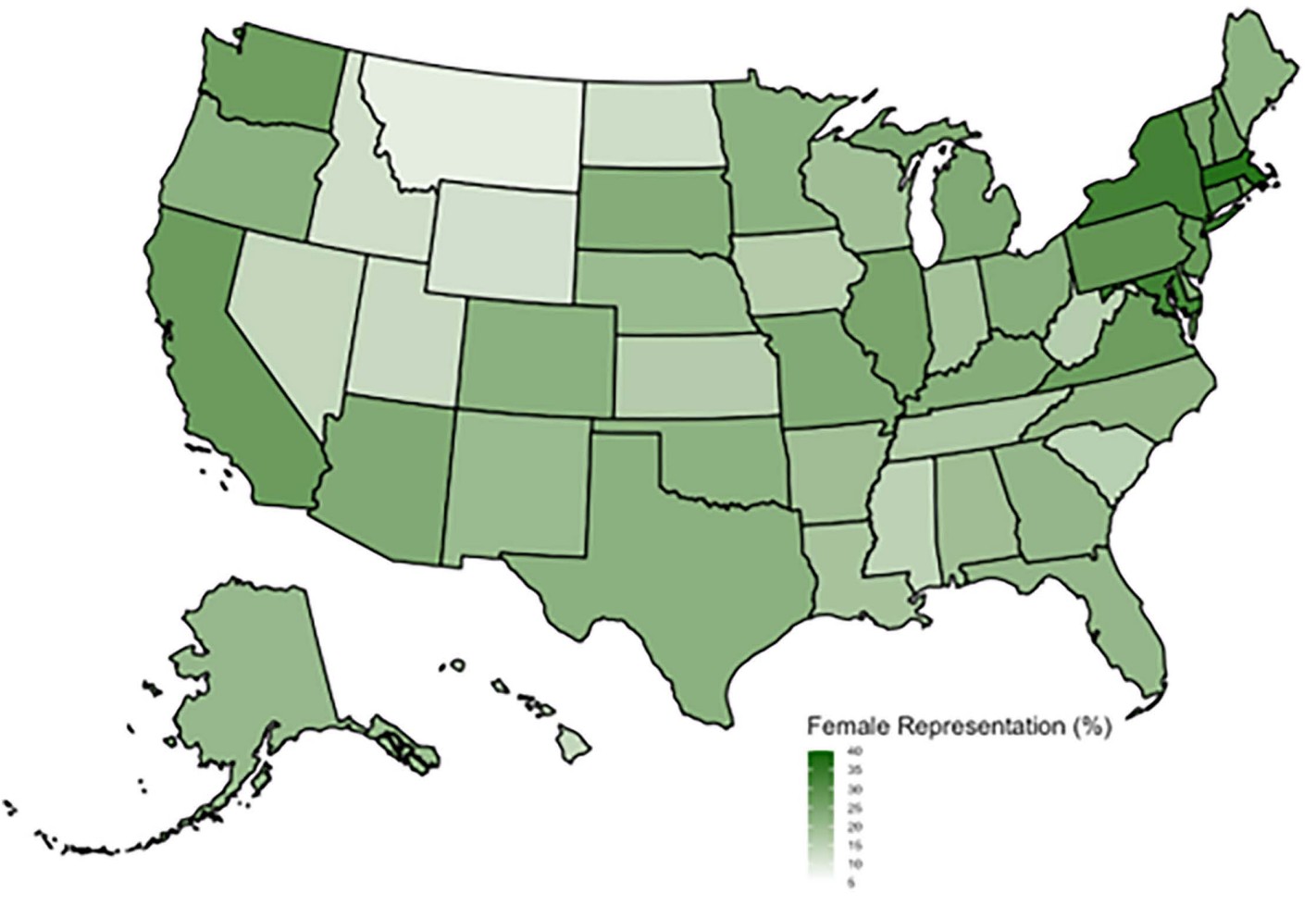

**Fig 1. Heat map of female radiologist representation by state.**

**Table 2. Medicare annual charges, payments, and practice volume metrics by gender.**

|  | Females | Males | P-value |
|---|---|---|---|
| **Charges and payments*** | **Mean (SE)** |  |  |
| Total submitted charges | $1,472,306 (9,066) | $1,928,044 (6,363) | <.001 |
| Total payments | $301,931 (1,810) | $352,604 (1,164) | <.001 |
| Payment-to-charge ratio | 0.23 (0.0006) | 0.21 (0.0003) | <.001 |
| Payment-per-Service | $36 (0.13) | $31 (0.06) | <.001 |
| **Practice volume** | **Median (IQR)** |  |  |
| Total services | 6,660 (3,168−11,990) | 9,859 (4,938−16,575) | <.001 |
| Total beneficiaries | 5,441 (2,694−9,159) | 7,678 (3,958−12,231) | <.001 |
| Unique codes | 20 (10-37) | 39 (21-55) | <.001 |

*Adjusted by Consumer Product Index (CPI).

**Table 3. Medicare payments by demographic characteristic and gender.**

Medicare payments, services, and payment per service by gender

| | Females | | | Males | | | Female-to-male ratio | | |
|---|---|---|---|---|---|---|---|---|---|
| | Average payments per radiologist | Average services per radiologist | Payment per service | Average payments per radiologist | Average services per radiologist | Payment per service | Average payments per radiologist | Average services per radiologist | Payment per service |
| Overall | $301,931 | 6600 | $36.26 | $352,604 | 9859 | $30.71 | 0.86 | 0.67 | 1.18 |
| Academic status | | | | | | | | | |
| Academic | $214,777 | 7521 | $32.99 | $241,078 | 9553 | $30.96 | 0.89 | 0.79 | 1.07 |
| Nonacademic | $321,905 | 9725 | $36.50 | $369,876 | 13483 | $29.68 | 0.87 | 0.72 | 1.23 |
| Practice rurality | | | | | | | | | |
| Urban | $293,304 | 9068 | $36.14 | $344,718 | 12517 | $30.90 | 0.85 | 0.72 | 1.17 |
| Rural | $285,182 | 10483 | $28.90 | $366,273 | 14243 | $26.73 | 0.78 | 0.74 | 1.08 |
| Years in practice | | | | | | | | | |
| <10 | $162,896 | 5543 | $30.81 | $181,277 | 6970 | $27.07 | 0.9 | 0.8 | 1.14 |
| 10-24 | $297,113 | 9316 | $36.01 | $361,802 | 13257 | $30.56 | 0.82 | 0.7 | 1.18 |
| ≥25 | $333,339 | 10141 | $37.70 | $380,922 | 13827 | $31.54 | 0.88 | 0.73 | 1.2 |
| Group practice size | | | | | | | | | |
| <10 members | $347,853 | 9432 | $46.93 | $437,750 | 14613 | $34.83 | 0.79 | 0.65 | 1.35 |
| 10–49 member | $356,377 | 10890 | $35.71 | $395,448 | 14493 | $29.49 | 0.9 | 0.75 | 1.21 |
| 50–99 memers | $347,673 | 10887 | $35.51 | $373,203 | 14110 | $29.01 | 0.93 | 0.77 | 1.22 |
| ≥100 members | $250,529 | 8056 | $34.85 | $287,700 | 10767 | $30.37 | 0.87 | 0.75 | 1.15 |
| Geographic region | | | | | | | | | |
| Midwest | $245,397 | 8837 | $30.56 | $279,083 | 10774 | $28.38 | 0.88 | 0.82 | 1.08 |
| Northeast | $281,299 | 8261 | $39.56 | $341,685 | 11436 | $34.79 | 0.82 | 0.72 | 1.14 |
| South | $305,434 | 9676 | $35.40 | $356,204 | 13537 | $29.47 | 0.86 | 0.71 | 1.2 |
| West | $270,707 | 7790 | $36.76 | $310,045 | 11325 | $30.50 | 0.87 | 0.69 | 1.21 |
| Subspecialty | | | | | | | | | |
| Abdominal | $256,901 | 9002 | $34.23 | $339,337 | 12421 | $33.25 | 0.76 | 0.72 | 1.03 |
| Breast | $348,755 | 7289 | $46.62 | $409,468 | 8815 | $46.41 | 0.85 | 0.83 | 1 |
| Cardiothoracic | $142,146 | 6021 | $16.74 | $153,356 | 7185 | $16.03 | 0.93 | 0.84 | 1.04 |
| Musculoskeletal | $208,936 | 7441 | $36.00 | $277,424 | 8861 | $38.40 | 0.75 | 0.84 | 0.94 |
| Neuroradiology | $237,174 | 6523 | $39.91 | $310,432 | 8631 | $39.91 | 0.76 | 0.76 | 1 |
| Generalists | $326,939 | 11130 | $29.07 | $375,572 | 13391 | $27.56 | 0.87 | 0.83 | 1.05 |

practices and work as subspecialists particularly in breast, cardiothoracic, and abdominal imaging. Further, total Medicare payments received by females are less than males as are the services they rendered to Medicare patients, but on a per service basis, female radiologists' Medicare payments per services were 18% more than males.

We found that female radiologists were disproportionately in academic radiology at 34.2%. This is consistent with previous studies of academic radiologists published in 2016 and 2017, which ranged from 29–34% [24,25]. The share of females in academic radiology has risen over time from 11.5% in 1978 to 28.1% in 2013 [26]. We found that by years of practice, females represented 25.5%, 28.0%, and 22.2% of radiologists with 1–9, 10–24 and ≥25 years of practice, respectively. While these shares indicate some increase in female representation over time because the share was lowest

**Changes over Time in Gender-Based Differences**

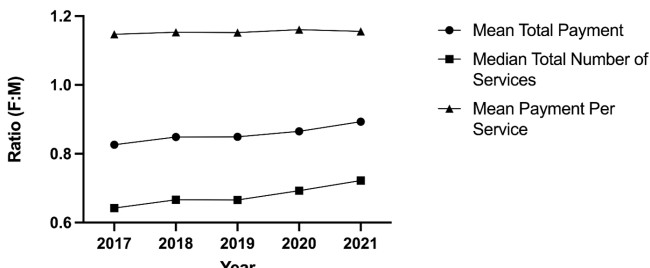

**Fig 2. Change over time in median number of services, mean payment per service and mean total payment to females compared to male radiologists.**

for those with ≥25 years of practice, we did not observe those with 1–9 years of practice having the highest female representation. Hence, if such a trend continues, female representation in radiology may plateau.

We found that female radiologists were disproportionately subspecialized, most commonly working in breast, cardio-thoracic, and abdominal imaging. This is consistent with a study that found female radiologists were more likely to only practice in one specialty [27]. Interestingly, while we found that females were the least represented in musculoskeletal imaging, this likely does not translate to females being less represented in academic musculoskeletal radiology as a previous study found that 30.7% of these faculty were females [28]. We found that females were more likely in urban areas, which is consistent with the prior study [29]. This combined with females' higher propensity to be employees and disproportionate representation in academic practices is consistent with our finding that females are disproportionately in large practices [29].

An array of similar studies found the existence of a gender-based payment difference in other specialties, such as but not limited to radiation oncology, ophthalmology, otolaryngologists, cardiology, and neurosurgery [9–13]. Considering only total Medicare payments, we found that female radiologists' payments from Medicare were 86% that of males. While some studies have found a gender pay difference for radiologists, our results are only representative of payments from Medicare not all sources. For example, a 2021 study found the female physicians' income was 12% less than males after adjusting for covariates. [14] This same study found that females Medicare reimbursement was 52% less than males [14]. This may suggest that female radiologists may derive a relatively larger proportion of their income from non-Medicare sources such as commercial insurance or institutional compensation models, though this remains a hypothesis and was not directly tested in our analysis. Consequently, these findings represent differences in Medicare FFS reimbursement and should not be interpreted as a measure of total physician compensation.

Furthermore, our multivariable, regression-adjusted results offer deeper insight into the observed gender-based differences in radiology practice. Even after controlling for service volume, practice setting, years of experience, and subspecialty, female radiologists continued to submit lower annual charges yet received higher Medicare payments. This divergence indicates that compensation differences are not solely attributable to differences in workload or access to billing opportunities, but may reflect underlying variations in clinical roles, coding patterns, or the relative value of services performed. Importantly, payment per service is influenced by differences in subspecialty, modality mix, and geographic adjustment factors. The higher adjusted payments observed among female radiologists may suggest a focus on more complex or evaluative services that yield greater reimbursement per encounter. Taken together, these findings underscore the importance of examining not only how much radiologists work, but also the nature and structure of the work they perform, when evaluating gender-based disparities in compensation. Surveys have also found that female radiologists earned less than males with varying estimates of differences with one study finding a 23% difference [29,30]. A recent

study of academic U.S. radiology faculty using AAMC data found that female radiologist's earnings were 3–6% lower than their male counterparts [4,7]. While these studies have all adjusted for covariates, the greatest differences are derived from surveys of radiologists generally, with smaller differences from surveys of like radiologists, such as surveys of only academic radiologists, and no difference for the study of reported salaries of all academic radiologists of the institutions included in the study. This may suggest that covariates in some studies insufficiently capture differences across radiologists as estimated earnings differences decrease or become insignificant as the samples because more alike. This possibility is consistent with our finding that once Medicare payments were normalized by differences in the number of services rendered, females were paid 18% more per service. Concluding that females are paid 18% more than males is misleading because Medicare payments are set by the Medicare Physician Fee Schedule. Hence, differences in payments per service can only stem from differences in the mix and count of specific services rendered, the share of services that include the technical component in addition to the professional component, and reimbursement differences associated with the Geographic Practice Cost Index. Such differences would be associated with how female and male radiologists sort themselves across practice type, location, and size, but principally by subspecialty. We found that for radiologists in the same subspecialty, there was generally gender parity in payments per service, which is logical given the Medicare Physician Fee Schedule.

## Limitations

Our study has limitations. First, the data are limited to Medicare billing data from 2017–2021, which excludes data for physicians who do not accept Medicare; however, only fewer than 0.1% of radiologists do not accept Medicare [31]. The uniformity of the Medicare data was crucial to this study's objective to examine payment trends among diagnostic radiologists submitting Medicare fee-for-service claims. Second, since services provided to patients are covered by other payors (i.e., commercial insurers, Medicaid, and Medicare Advantage), it is possible that characterization of radiologists' subspecialties based on Medicare fee-for-service data alone, may mischaracterize the subspecialty in some instances. Further, we used a previously established method (using wRVUs and NITOS codes) to determine radiologists' subspecialties based on services rendered, not fellowship training [19–21]. Lastly, while this study focused on Medicare payments to physicians, factors including practice type (private vs employed) and hospital policies may differentially affect reimbursement and the amount of actual dollars that end up in doctors' take-home salaries and compensation. Additionally, it is important to note that data on working hours or full-time equivalent (FTE) status were not available and could influence payment differences. Furthermore, gender was captured as a self-reported binary variable within the CMS/NPPES datasets; consequently, this study is limited by the lack of representation for non-binary gender identities.

In conclusion, we found that relative to the one-quarter of radiologists overall who are female, female radiologists are disproportionately in academic, urban, and large practices and work as subspecialists particularly in breast, cardiothoracic, and abdominal imaging. Further, while total Medicare payments received by females are less than males, payments per service rendered were higher for female radiologists than for males overall, but were comparable between genders for the same subspecialty.

### Key points

1. Female radiologists are better represented in academic radiology, practice in urban areas, in larger practices and in certain subspecialties.

2. On average, female radiologists submitted charges that were 76% of that submitted by males and were paid by Medicare 86% of what male radiologists were paid.

3. There was generally parity between female and male radiologists in payments per service for those in the same subspecialty

## Author contributions

**Conceptualization:** Ajay Malhotra, Seyedmehdi Payabvash, David Seidenwurm, Dheeraj Gandhi.

**Data curation:** Ajay Malhotra, Chris J. Lee, Mihir Khunte, Raj Moily, Dheeman Futela.

**Formal analysis:** Ajay Malhotra, Chris J. Lee, Mihir Khunte, Raj Moily, Dheeman Futela.

**Investigation:** Ajay Malhotra, Chris J. Lee, Mihir Khunte, Raj Moily, Dheeman Futela, Seyedmehdi Payabvash, David Seidenwurm, Dheeraj Gandhi.

**Methodology:** Ajay Malhotra, Mihir Khunte, Raj Moily, Seyedmehdi Payabvash.

**Project administration:** Ajay Malhotra.

**Supervision:** Ajay Malhotra, Seyedmehdi Payabvash, David Seidenwurm, Dheeraj Gandhi.

**Validation:** Chris J. Lee, Mihir Khunte.

**Visualization:** Ajay Malhotra, Chris J. Lee, Mihir Khunte, Raj Moily, Dheeman Futela.

**Writing – original draft:** Ajay Malhotra, Chris J. Lee, Mihir Khunte, Raj Moily, Dheeman Futela.

**Writing – review & editing:** Ajay Malhotra, Chris J. Lee, Mihir Khunte, Raj Moily, Dheeman Futela, Seyedmehdi Payabvash, David Seidenwurm, Dheeraj Gandhi.

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
