## [Decision Letter · Decision Letter 0]

6 Jun 2025

PONE-D-25-20867Gender Differences in Provider Practice Characteristics and Medicare Payment & Services Among Diagnostic RadiologistsPLOS ONE

Dear Dr. Malhotra,

Thank you for submitting your manuscript to PLOS ONE. After careful consideration, we feel that it has merit but does not fully meet PLOS ONE’s publication criteria as it currently stands. Therefore, we invite you to submit a revised version of the manuscript that addresses the points raised during the review process.

We look forward to receiving your revised manuscript.

Kind regards,

Muhammad Muntazir Mehdi Khan, M.B.B.S.

Academic Editor

PLOS ONE

Journal Requirements:

3. Please amend the manuscript submission data (via Edit Submission) to include author Seyedmehdi Payabvash.

4. Please amend your authorship list in your manuscript file to include author Sam Payabvash.

Reviewers' comments:

Reviewer's Responses to Questions

**Comments to the Author**

1. Is the manuscript technically sound, and do the data support the conclusions?

Reviewer #1: Yes

Reviewer #2: Partly

2. Has the statistical analysis been performed appropriately and rigorously? 

Reviewer #1: Yes

Reviewer #2: No

3. Have the authors made all data underlying the findings in their manuscript fully available?

Reviewer #1: No

Reviewer #2: Yes

4. Is the manuscript presented in an intelligible fashion and written in standard English?

Reviewer #1: Yes

Reviewer #2: Yes

5. Review Comments to the Author

Reviewer #1: The manuscript presents an original and timely analysis of gender-based disparities in practice characteristics and Medicare payments among diagnostic radiologists. The topic is highly relevant, and the analysis is based on a large, publicly available dataset, with appropriate statistical comparisons. The manuscript is technically sound, and the conclusions are well-supported by the data. However, I recommend minor revisions to address key clarifications and align with PLOS ONE’s standards for reproducibility and transparency.

Abstract: On Line 28, correct “differences among radiologists is poorly understood” to “are poorly understood.” On Lines 57–60, clarify the payment comparison: “female radiologists received 86% of the total Medicare payments received by male radiologists.” Remove space before semicolons in statistical reporting (e.g., Line 55: “9,859 ;” → “9,859;”). This line is a little confusing, please fix this.

Methods (Lines 91–141): The methodology is appropriate, but a few key variable definitions are missing. Please clarify:

How gender was assigned (e.g., from NPPES or another linked source) — not specified in current version.

How academic vs. non-academic status was defined (Line 112 onward).

How urban vs. rural practice was classified (Line 114).

How practice size was measured (Line 116).

Line 120 contains a typo: correct “breath of practice” to “breadth of practice.” On Line 133, clarify whether a two-tailed alpha level (e.g., α = 0.05) was used and if any R packages were applied in analysis.

Results (Lines 142–191): The results are clearly presented and statistically robust. To improve clarity:

On Lines 158–161, rephrase for interpretability: “34.2% of academic radiologists were female, compared to 22.6% in non-academic settings.”

On Line 164, clarify that 64.9% refers to female representation within breast imaging.

On Lines 142–145, specify how p-values for time trends were calculated (e.g., test for trend vs simple comparison).

For statements like “per-service payments were similar within subspecialties” (Line 172), consider adding whether statistical testing confirmed this or clarify as an observation.

Discussion (Lines 192–247): Well-structured and grounded in existing literature. On Line 193, fix heading typo: “DICUSSION” → “DISCUSSION.” Line 184 contains a grammatical error: “the a prior study” → “a prior study.” On Line 159, temper speculation: replace “will plateau” with “may plateau.” Similarly, Line 107–110 speculates about non-Medicare income sources — please clearly frame this as a hypothesis. Consider briefly discussing implications or areas for future research (e.g., understanding drivers of volume differences).

Limitations (Lines 248–266): Strong and transparent. On Line 252, rephrase “only <0.1%” to “fewer than 0.1%.” Consider noting that working hours or FTE status were not available and could influence payment differences. Also mention that multivariable adjustment was not performed.

Conclusion (Lines 263–266): Appropriate and data-aligned. Line 264 may read better as: “payments per service were higher for female radiologists than for males overall, but were comparable between genders within the same subspecialty.”

Data Availability (Lines 93–95, 267–268): Current statement (“CMS Public files”) is insufficient. Please expand to include dataset name, years used, and a direct URL (e.g., “https://data.cms.gov/...”).

Minor Style Issues: Maintain consistent terminology (e.g., “female/male radiologists”), fix small typos, and add table/figure cross-references where appropriate in Results.

In summary, this is a strong manuscript that requires minor revisions to address transparency, clarify definitions, and polish a few points of language and formatting. Once addressed, it will meet the publication standards of PLOS ONE.

Reviewer #2: In the study by Malhotra et al., the authors sought to compare Medicare payments between female and male diagnostic radiologists. Overall, this is an interesting study and is well written. I have a major concern:

Why did the authors not conduct a multivariate regression to see an independent association between and medicare payments? From the results, it is seen that female radiologists are more likely to be in academic practice compared to male radiologists. The practice type could be a confounder/ effect modifier and therefore, we cannot draw conclusions without multivariate analysis.

6. PLOS authors have the option to publish the peer review history of their article (what does this mean?). If published, this will include your full peer review and any attached files.

Reviewer #1: **Yes:** N/A

Reviewer #2: No

---

## [Author Response · Author response to Decision Letter 1]

10 Jul 2025

Reviewers' comments:

Reviewer's Responses to Questions

Comments to the Author

1. Is the manuscript technically sound, and do the data support the conclusions?

Reviewer #1: Yes

Reviewer #2: Partly

Response: We have added to the analysis as suggested.

2. Has the statistical analysis been performed appropriately and rigorously?

Reviewer #1: Yes

Reviewer #2: No

Response: We have added to the statistical analysis as suggested and performed multivariable regression- this has been added to the Methods, Results and Discussion sections.

3. Have the authors made all data underlying the findings in their manuscript fully available?

Reviewer #1: No

Reviewer #2: Yes

Response: We have now added the Data Availability Statement and clarified that the data is publicly available and provided the source.

4. Is the manuscript presented in an intelligible fashion and written in standard English?

Reviewer #1: Yes

Reviewer #2: Yes

5. Review Comments to the Author

Reviewer #1: The manuscript presents an original and timely analysis of gender-based disparities in practice characteristics and Medicare payments among diagnostic radiologists. The topic is highly relevant, and the analysis is based on a large, publicly available dataset, with appropriate statistical comparisons. The manuscript is technically sound, and the conclusions are well-supported by the data. However, I recommend minor revisions to address key clarifications and align with PLOS ONE’s standards for reproducibility and transparency.

Abstract: On Line 28, correct “differences among radiologists is poorly understood” to “are poorly understood.” On Lines 57–60, clarify the payment comparison: “female radiologists received 86% of the total Medicare payments received by male radiologists.” Remove space before semicolons in statistical reporting (e.g., Line 55: “9,859 ;” → “9,859;”). This line is a little confusing, please fix this.

Response: We have now incorporated the suggestions and made corrections.

Methods (Lines 91–141): The methodology is appropriate, but a few key variable definitions are missing. Please clarify:

How gender was assigned (e.g., from NPPES or another linked source) — not specified in current version.

How academic vs. non-academic status was defined (Line 112 onward).

How urban vs. rural practice was classified (Line 114).

How practice size was measured (Line 116).

Response: We have now incorporated the suggestions and clarified the definition sources.

Line 120 contains a typo: correct “breath of practice” to “breadth of practice.” On Line 133, clarify whether a two-tailed alpha level (e.g., α = 0.05) was used and if any R packages were applied in analysis.

Results (Lines 142–191): The results are clearly presented and statistically robust. To improve clarity:

On Lines 158–161, rephrase for interpretability: “34.2% of academic radiologists were female, compared to 22.6% in non-academic settings.”

On Line 164, clarify that 64.9% refers to female representation within breast imaging.

Response: We have now incorporated the suggestions and made corrections.

On Lines 142–145, specify how p-values for time trends were calculated (e.g., test for trend vs simple comparison).

Response: We have now incorporated the suggestion and added to Methods and Results. Trends in payment amounts over time were determined with a time-series analysis with a two-tailed alpha level (α = 0.05).

For statements like “per-service payments were similar within subspecialties” (Line 172), consider adding whether statistical testing confirmed this or clarify as an observation.

Response: We have now incorporated the suggestion.

Discussion (Lines 192–247): Well-structured and grounded in existing literature. On Line 193, fix heading typo: “DICUSSION” → “DISCUSSION.” Line 184 contains a grammatical error: “the a prior study” → “a prior study.” On Line 159, temper speculation: replace “will plateau” with “may plateau.” Similarly, Line 107–110 speculates about non-Medicare income sources — please clearly frame this as a hypothesis. Consider briefly discussing implications or areas for future research (e.g., understanding drivers of volume differences).

Response: We have now incorporated the suggestions and made corrections.

Limitations (Lines 248–266): Strong and transparent. On Line 252, rephrase “only <0.1%” to “fewer than 0.1%.” Consider noting that working hours or FTE status were not available and could influence payment differences. Also mention that multivariable adjustment was not performed.

Response: We have now incorporated the suggestions and made corrections.

Conclusion (Lines 263–266): Appropriate and data-aligned. Line 264 may read better as: “payments per service were higher for female radiologists than for males overall, but were comparable between genders within the same subspecialty.”

Response: We have now incorporated the suggestions and made corrections.

Data Availability (Lines 93–95, 267–268): Current statement (“CMS Public files”) is insufficient. Please expand to include dataset name, years used, and a direct URL (e.g., “https://data.cms.gov/...”).

Response: We have now incorporated the suggestions and made corrections.

Minor Style Issues: Maintain consistent terminology (e.g., “female/male radiologists”), fix small typos, and add table/figure cross-references where appropriate in Results.

In summary, this is a strong manuscript that requires minor revisions to address transparency, clarify definitions, and polish a few points of language and formatting. Once addressed, it will meet the publication standards of PLOS ONE.

Reviewer #2: In the study by Malhotra et al., the authors sought to compare Medicare payments between female and male diagnostic radiologists. Overall, this is an interesting study and is well written. I have a major concern:

Why did the authors not conduct a multivariate regression to see an independent association between and medicare payments? From the results, it is seen that female radiologists are more likely to be in academic practice compared to male radiologists. The practice type could be a confounder/ effect modifier and therefore, we cannot draw conclusions without multivariate analysis.

Response: We have now incorporated the suggestions and performed multivariate regression- this has been added to the Methods, Results and Discussion sections.

6. PLOS authors have the option to publish the peer review history of their article (what does this mean?). If published, this will include your full peer review and any attached files.

Do you want your identity to be public for this peer review? For information about this choice, including consent withdrawal, please see our Privacy Policy.

Reviewer #1: Yes: N/A

Reviewer #2: No

---

## [Decision Letter · Decision Letter 1]

10 Mar 2026

PONE-D-25-20867R1Gender Differences in Provider Practice Characteristics and Medicare Payment & Services Among Diagnostic RadiologistsPLOS One

Dear Dr. Malhotra,

Thank you for submitting your revised manuscript to PLOS ONE. Several issues have been addressed, but other ones have been raised by the reviewers in the last review round. Therefore, we invite you to submit a further revised version of the manuscript that addresses the latest points.

We look forward to receiving your revised manuscript.

Kind regards,

Lorenzo Faggioni, M.D., Ph.D.

Academic Editor

PLOS One

Journal Requirements:

Reviewers' comments:

Reviewer's Responses to Questions

**Comments to the Author**

1. If the authors have adequately addressed your comments raised in a previous round of review and you feel that this manuscript is now acceptable for publication, you may indicate that here to bypass the “Comments to the Author” section, enter your conflict of interest statement in the “Confidential to Editor” section, and submit your "Accept" recommendation.

Reviewer #3: (No Response)

Reviewer #4: (No Response)

2. Is the manuscript technically sound, and do the data support the conclusions?

Reviewer #3: Partly

Reviewer #4: Yes

3. Has the statistical analysis been performed appropriately and rigorously? 

Reviewer #3: No

Reviewer #4: Yes

4. Have the authors made all data underlying the findings in their manuscript fully available?

Reviewer #3: Yes

Reviewer #4: Yes

5. Is the manuscript presented in an intelligible fashion and written in standard English?

Reviewer #3: Yes

Reviewer #4: Yes

6. Review Comments to the Author

Reviewer #3: Doctors --

OVERALL

This manuscript addresses an interesting, policy-relevant question; the CMS Physician/Other Practitioners public files are a reasonable source for a descriptive look at radiology workload and Medicare payments.

Your headline findings: women are ~25% of diagnostic radiologists; more likely to be academic, urban, in large groups, and in certain subspecialties (notably breast); women have lower total Medicare payments but higher payment per service overall, with parity within the same subspecialty. All of that is clearly stated.

RECOMMENDATIONS

1. Adjust beyond bivariate tests.

Right now, means/medians are compared with t-tests/Wilcoxon. That’s not enough for a question this confounded (years in practice, region, urbanicity, practice size, subspecialty mix, year). Add multivariable models (e.g., panel OLS/GLM with radiologist and year fixed effects or at least covariate adjustment with clustered SEs by NPI).

Report adjusted differences and CIs.

2. Decompose “payment per service.”

Your own discussion concedes this metric reflects code mix, professional vs global billing, facility vs non-facility, and regional GPCI, not “pay” per se. Do a mix-adjustment: (i) show the top CPT/HCPCS distributions by sex; (ii) reweight to a common code mix; (iii) report a case-mix–adjusted payment per service. That will test whether the 18% higher female PPS is mix, geography, or true per-code differences (which should be zero under the fee schedule).

Also break out professional-only vs global where feasible.

3. Subspecialty classification transparency.

You infer subspecialty by >50% wRVUs using NITOS modality/body-region mapping. Clever idea — recommend to document it fully in the Methods (code lists, thresholds, sensitivity with 60%/70% cutoffs). Right now the summary references prior validation but your implementation details (exact crosswalks) need to be reproducible. Deposit the mapping.

4. Define “academic vs non-academic,” “urban vs rural,” and “years in practice.”

You cite practice location and NPPES for “years in practice,” but the operational definitions aren’t spelled out (RUCA vs MSA? Years since NPI enumeration != years in practice). Add precise algorithms and references.

5. Repeated measures / clustering.

Are observations pooled across 2017–2021 at the physician-year level? If so, your tests must account for within-physician correlation across years (clustered SEs) and secular trends (year fixed effects). The time-trend figure suggests a panel; analyze it as such.

6. Tables contain obvious inconsistencies.

In Table 1, the “All years” column shows tiny denominators (e.g., “Academic 34.2% (221/948)”) that don’t match the reported overall N (33,029). Audit and correct these lines before anything else. Similar spot-checks across Table 3 are prudent.

7. Terminology and source for “gender.”

State explicitly how “gender” is obtained in CMS files (binary sex field, self-report, inference?). Use consistent terminology and acknowledge limitations (non-binary not captured).

8. Scope the inference carefully.

You study FFS Medicare claims only. Don’t imply salary conclusions or “pay gaps”; these are Medicare reimbursements, not total compensation, and exclude Medicare Advantage/commercial. You mention this in Limitations—bring that caution forward into the Abstract and Conclusions.

9. Data/code availability.

In the spirit of replicability: provide exact dataset names/years/URLs and deposit all code and crosswalks (wRVU/NITOS subspecialty assignment, urban/rural rules, academic flag) in a public repo. The current data statement should point precisely to those sources.

MINOR NOTES

- Use rates per radiologist-year (or per 1,000 services) in the text instead of raw totals; keep totals to the tables.

- Report both means and medians for skewed financial variables.

- Tighten typos (e.g., “DICUSSION”), and standardize style in tables/figures.

ASSESSMENT

Publishable as a descriptive claims analysis once you fix the table errors, add proper adjustment/mix-decomposition, define classification rules, and scope claims to Medicare reimbursements (not income).

The current unadjusted comparisons over-interpret the 18% PPS difference.

Reviewer #4: Thank you for the opportunity to review this manuscript. The manuscript can benefit from the following revisions:

1) The manuscript does not define how “academic” radiologists were identified. Was this based on practice setting codes, affiliation with teaching hospitals, or another CMS variable? Please specify in the Methods.

2) The result that females receive 18% higher payment per service is interesting.

a) Could it be that female radiologists perform more complex/higher RVU studies?

b) Are they more likely to bill for the technical component (e.g., in breast imaging)?

c) Is there a geographic concentration in higher-GPCI regions?

A brief discussion should be added for these points.

3) The use of wRVU-based assignment is validated, but it remains an approximation.

4) The authors should explain that why they chose only medicare population because the real pay disparity lies in the commercial payer rates.

5) The authors note that female representation is not increasing among newer radiologists (1–9 years) compared to mid-career (10–24 years), suggesting a potential plateau. This is an important observation and should be discussed in the context of national workforce trends and potential barriers to entry.

7. PLOS authors have the option to publish the peer review history of their article (what does this mean?). If published, this will include your full peer review and any attached files.

Reviewer #3: No

Reviewer #4: No

---

## [Author Response · Author response to Decision Letter 2]

20 Apr 2026

6. Review Comments to the Author

Reviewer #3: Doctors --

OVERALL

This manuscript addresses an interesting, policy-relevant question; the CMS Physician/Other Practitioners public files are a reasonable source for a descriptive look at radiology workload and Medicare payments.

Your headline findings: women are ~25% of diagnostic radiologists; more likely to be academic, urban, in large groups, and in certain subspecialties (notably breast); women have lower total Medicare payments but higher payment per service overall, with parity within the same subspecialty. All of that is clearly stated.

RECOMMENDATIONS

1. Adjust beyond bivariate tests.

Right now, means/medians are compared with t-tests/Wilcoxon. That’s not enough for a question this confounded (years in practice, region, urbanicity, practice size, subspecialty mix, year). Add multivariable models (e.g., panel OLS/GLM with radiologist and year fixed effects or at least covariate adjustment with clustered SEs by NPI).

Report adjusted differences and CIs.

Response: Thank you for your comment. In our analysis, we aggregated the data at the NPI level for each individual calendar year, such that each observation represents a physician-year. While this structure allows for repeated observations of the same physician across years, practice patterns may vary meaningfully over time (e.g., due to sabbaticals, transitions to part-time work, or retirement). Because these factors cannot be reliably captured in the dataset, we did not consider it appropriate to pool observations across years at the physician level.

Also, we do not observe substantial year-to-year variations, and the calendar year was not considered a significant confounder. Accordingly, we utilized unadjusted comparisons (t-tests and Wilcoxon rank-sum tests) to evaluate differences between groups.

2. Decompose “payment per service.”

Your own discussion concedes this metric reflects code mix, professional vs global billing, facility vs non-facility, and regional GPCI, not “pay” per se. Do a mix-adjustment: (i) show the top CPT/HCPCS distributions by sex; (ii) reweight to a common code mix; (iii) report a case-mix–adjusted payment per service. That will test whether the 18% higher female PPS is mix, geography, or true per-code differences (which should be zero under the fee schedule).

Also break out professional-only vs global where feasible.

Response: We agree that the observed difference in payment per service likely reflects variations in code mix, billing composition, and geographic factors. While a formal case-mix adjustment falls outside the macro-level scope of our study, Table 1 outlines broad distributional differences in practice composition by gender. To address this important point, we have added a sentence to the Discussion clarifying that this metric is driven by differences in subspecialty and modality mix.

3. Subspecialty classification transparency.

You infer subspecialty by >50% wRVUs using NITOS modality/body-region mapping. Clever idea — recommend to document it fully in the Methods (code lists, thresholds, sensitivity with 60%/70% cutoffs). Right now the summary references prior validation but your implementation details (exact crosswalks) need to be reproducible. Deposit the mapping.

Response: Thank you for your comment. We will document it in the Methods. This threshold and classification strategy were derived from the methodology described by Rosenkrantz et al. (J Am Coll Radiol, 2017), which validated a claims-based approach to subspecialty assignment using Medicare data. We adapted their approach by applying the same >50% wRVU threshold to assign each radiologist a single dominant subspecialty based on their distribution of billed services.

Rosenkrantz AB, Wang W, Hughes DR, Ginocchio LA, Rosman DA, Duszak R Jr. Academic Radiologist Subspecialty Identification Using a Novel Claims-Based Classification System. AJR Am J Roentgenol. 2017 Jun;208(6):1249-1255. doi: 10.2214/AJR.16.17323. Epub 2017 Mar 16. Erratum in: AJR Am J Roentgenol. 2017 Aug;209(2):472. doi: 10.2214/AJR.17.18609. PMID: 28301213.

Rosenkrantz AB, Wang W, Bodapati S, Hughes DR, Duszak R Jr. Private Practice Radiologist Subspecialty Classification Using Medicare Claims. J Am Coll Radiol. 2017 Nov;14(11):1419-1425. doi: 10.1016/j.jacr.2017.04.025. Epub 2017 Jun 30. PMID: 28673776.

4. Define “academic vs non-academic,” “urban vs rural,” and “years in practice.”

You cite practice location and NPPES for “years in practice,” but the operational definitions aren’t spelled out (RUCA vs MSA? Years since NPI enumeration != years in practice). Add precise algorithms and references.

Response: Thank you for your comment. We have clarified the operational definitions in the Methods section. The academic status of practices associated with radiologists in the CMS files was obtained from the Harvey L. Neiman Health Policy Institute. Per the methodology provided by the institute, practices were classified as academic if their names contained terms such as “university,” “faculty,” “college,” or “school,” and were further validated through manual review against the list of diagnostic radiology residency programs provided by the AAMC. Urban versus rural status was determined using Rural-Urban Commuting Area (RUCA) codes based on practice ZIP code. “Years in practice” was derived from the physician’s medical school graduation year as reported in the publicly available National Plan & Provider Enumeration System (NPPES) database and linked via NPI.

5. Repeated measures / clustering.

Are observations pooled across 2017–2021 at the physician-year level? If so, your tests must account for within-physician correlation across years (clustered SEs) and secular trends (year fixed effects). The time-trend figure suggests a panel; analyze it as such.

Response: Thank you for this comment. Our dataset is structured at the physician-year level, with each observation representing a unique NPI in a given calendar year. While this allows for repeated observations of the same physician across years, we did not model the data as a longitudinal panel. This decision was based on the potential for meaningful year-to-year variation in individual practice patterns (e.g., sabbaticals, transitions to part-time work, or retirement), which cannot be reliably captured in the dataset.

Also, we do not observe substantial year-to-year variations, and the calendar year was not considered a significant confounder. Accordingly, we utilized unadjusted comparisons (t-tests and Wilcoxon rank-sum tests) to evaluate differences between groups.

6. Tables contain obvious inconsistencies.

In Table 1, the “All years” column shows tiny denominators (e.g., “Academic 34.2% (221/948)”) that don’t match the reported overall N (33,029). Audit and correct these lines before anything else. Similar spot-checks across Table 3 are prudent.

Response: The denominators in Table 1 reflect only physicians with available data for the specified variable (e.g., practice type. As such, we excluded analysis like “Academic vs Non Academic” for physicians who did not have data available.

7. Terminology and source for “gender.”

State explicitly how “gender” is obtained in CMS files (binary sex field, self-report, inference?). Use consistent terminology and acknowledge limitations (non-binary not captured).

Response: Thank you for your comment. Gender was obtained from the CMS dataset as a self-reported, binary variable (male/female). We have clarified this in the Methods section and now acknowledge this as a limitation, as non-binary gender identities are not captured in the dataset.

8. Scope the inference carefully.

You study FFS Medicare claims only. Don’t imply salary conclusions or “pay gaps”; these are Medicare reimbursements, not total compensation, and exclude Medicare Advantage/commercial. You mention this in Limitations—bring that caution forward into the Abstract and Conclusions.

Response: Thank you for your comment. We agree that our inferences should be carefully scoped to reflect that the analysis is based on FFS Medicare claims data and does not represent total physician compensation. We have detailed this in the limitations section and added a sentence about this in the discussion.

9. Data/code availability.

In the spirit of replicability: provide exact dataset names/years/URLs and deposit all code and crosswalks (wRVU/NITOS subspecialty assignment, urban/rural rules, academic flag) in a public repo. The current data statement should point precisely to those sources.

Response: Thank you for your comment. All data used in this study are publicly available, and specific references to the datasets, including the CMS Medicare Fee-for-Service Provider Utilization and Payment Data (2017–2021) and the National Plan & Provider Enumeration System (NPPES), are provided in the Methods section and references. The code is available from the authors upon request.

MINOR NOTES

- Use rates per radiologist-year (or per 1,000 services) in the text instead of raw totals; keep totals to the tables.

- Report both means and medians for skewed financial variables.

- Tighten typos (e.g., “DICUSSION”), and standardize style in tables/figures.

Thank you for your comment.

---

## [Editor Report · Decision Letter 2]

29 Apr 2026

Gender Differences in Provider Practice Characteristics and Medicare Payment & Services Among Diagnostic Radiologists

PONE-D-25-20867R2

Dear Dr. Malhotra,

We’re pleased to inform you that your manuscript has been judged scientifically suitable for publication and will be formally accepted for publication once it meets all outstanding technical requirements.

Kind regards,

Lorenzo Faggioni, M.D., Ph.D.

Academic Editor

PLOS One

---

## [Editor Report · Acceptance letter]

PONE-D-25-20867R2

PLOS One

Dear Dr. Malhotra,

I'm pleased to inform you that your manuscript has been deemed suitable for publication in PLOS One. Congratulations! Your manuscript is now being handed over to our production team.

Kind regards,

on behalf of

Dr. Lorenzo Faggioni

Academic Editor

PLOS One